# Gut Microbial Targets in Inflammatory Bowel Disease: Current Position and Future Developments

**DOI:** 10.3390/biomedicines13030716

**Published:** 2025-03-14

**Authors:** Naveen Sivakumar, Ashwin Krishnamoorthy, Harshita Ryali, Ramesh P. Arasaradnam

**Affiliations:** 1Institute of Precision Diagnostics & Translational Medicine, University Hospitals Coventry & Warwickshire, Coventry CV2 2DX, UK; r.arasaradnam@warwick.ac.uk; 2Warwick Medical School, University of Warwick, Coventry CV4 7AL, UK; ashwin.krishnamoorthy@warwick.ac.uk; 3Southmead Hospital, North Bristol Trust, Bristol BS10 5NB, UK; harshita.ryali@nbt.nhs.uk

**Keywords:** inflammatory bowel disease, microbiome, microbiota, faecal transplantation, gene therapy, nanoparticle

## Abstract

Inflammatory bowel disease (IBD) is a debilitating condition in which surgery is often seen as a last resort. However, this is associated with morbidity and, in some cases, mortality. There are emerging therapies that seek to better modulate the immune response of hosts with IBD. **Aims**: The main aim of this study is to focus on novel therapies and techniques studied in the last year that are non-surgical treatments of IBD. **Methods**: We looked at all the research between March 2024 and February 2025 detailing treatment in IBD and focused on the gut microbiome and gene therapy. **Results**: Novel therapies are gaining traction in safety and popularity. The results from some animal studies show promise and, with FDA approval, some probiotic therapies show optimistic research potential for future human trials. **Conclusions**: The research into the diagnostics and novel therapies available on the horizon for humans is very promising. Animal studies have shown potentially transferrable and safe therapies that can target specific sites of inflammation. Modulating the inflammatory response is a powerful therapy with what is shown to be a reasonably safe profile to build further research on.

## 1. Introduction

Inflammatory bowel disease (IBD) encompasses both Crohn’s disease (CD) and ulcerative colitis (UC)—two functionally and histologically distinct conditions that share the element of chronic inflammation within the gastrointestinal tract and give rise to an array of sequalae, as well as increasing the risk of colorectal cancer.

IBD can be difficult to treat. Both UC and CD are heterogenous diseases, with a wide spectrum of disease activities. While UC can be cured, it requires the full surgical removal of the large bowel to achieve this, often with the result of a stoma bag for the patient, which comes with its own set of problems. CD cannot be cured; thus, the therapeutic goal is to achieve remission (Figure 1). Therefore, due to the ongoing need for medications, hospital follow-up appointments, and lost workdays for patients, IBD can present a significant financial burden to health systems and to society.

The incidence of IBD is increasing worldwide, for reasons that are not entirely certain. Some postulate that the rising IBD incidence is due to factors associated with the Western lifestyle, such as Western diet and lack of exposure to microorganisms that potentially regulate the immune system.

In order to focus on the treatment of IBD, we need to touch upon the aetiology of the disease. The exact cause is unknown, but it is highly likely to be a combination of genetic and environmental factors that results in altered functioning of the immune system, leading to an “attack” of the body’s own gastrointestinal cells.

Therefore, the mainstay of the current medical management is related to suppressing the immune system. Various medications such as steroids, antimetabolite medication, and new “biologic” treatments (often monoclonal antibodies) that target certain biochemical pathways are used to suppress the action of the immune system. Unfortunately, this can lead to an array of unwanted side effects and toxicities. Furthermore, it is likely that there are a huge number of subsets of UC and CD, and these medications are currently not “personalised” to each individual’s metabolic processes.

An evolving area of scientific interest involves the gut microbiome, which is the community of microorganisms living in the human digestive tract. Specifically, there exists a complex interplay between gut microbiome species and cells of the gastrointestinal tract, and this is implicated in both the maintenance of physiological health and the development of disease. An imbalance in the gut microbiome, in which certain deleterious species of microorganisms are increased relative to the more “favourable” species, is known as gut dysbiosis. In fact, it has been shown that IBD bacterial families that exist in abundance do not co-exist unlike in healthy individuals [1]. There are also emerging data to suggest that gut dysbiosis is associated with not just IBD but also its complications such as colorectal cancer [2,3].

Modulation of the gut microbiome could reduce inflammation and thus improve symptoms, maintain remission, and reduce relapses in IBD without dependence on medications with significant side effects. To draw a parallel, cancer treatment is evolving rapidly with the tailoring of therapies towards individual genetic mutations and “individualisation” of treatment. It is possible that the key to similar advances in IBD treatment lies within gut microbial therapeutic targets.

In this article, we aim to review the current and emerging gut microbial targets in IBD and, specifically, to elucidate the future developments most likely to impact practice.

## 2. Overview of the Gut Microbiome and Its Implications in IBD

In a healthy physiological state, the balance between beneficial and detrimental microorganisms in the gut is maintained, with a particular preference for certain anti-inflammatory species such as Firmicutes and Bacteroides [4,5]. Some of these beneficial bacteria were noted to be reduced in individuals with disease compared with healthy individuals, whereas the reverse seemed to be true for some potentially harmful strains. For example, certain strains of mucolytic bacteria have been found more in individuals with Crohn’s and ulcerative colitis [6].

Such disruptions in the gut microbiome, dysbioses, may lead to IBD via inflammation, disruption of gut immune function, and the disruption of gut barrier function or short-chain fatty acid (SCFA) production (Figure 2). Foppa et al., 2024 [6] showed that the microbiota environment is linked to the pathogenesis of IBD via immune system change and microbiota composition. Recent reviews have even suggested that the gut microbiota can be used as a potential non-invasive biomarker for inflammatory bowel disease status and activity. This could change practice by using potentially cheaper and less invasive methods for disease surveillance. If checking the strains of bacteria in diseased individuals could accurately indicate disease activity or remission, then fewer endoscopic procedures would be required, and the potential risks to patients would be reduced.

Guo et al. [4] found that the bacteria *F. prausnitzii* is a specific marker for IBD as its prevalence was significantly lower than in IBS patients. They also found that, when considered in tandem with *E. coli*, *F. prausnitzii* could differentiate between colonic Crohn’s disease and extensive colitis, implying that combinations or compositions of bacteria within a gut microbiome may be a more useful biomarker for IBD than just one strain, thus showing us potential applications for studying the gut microbiome composition as a less invasive biomarker in the diagnosis of IBD, as well as differentiating it from IBS, which can be difficult in primary care or more ambiguous presentations.

Ocansey et al., 2023 [5] found that specific bacterial clusters activate NOD2 signalling pathways. These are part of the innate immune system and are involved in the detection and initial response to pathogens. It also has a gene identified as being susceptible to IBD. These findings were linked to changes in the patients’ gut microbiome, suggesting that the imbalances in the gut microbiota may contribute to the dysbiosis seen in IBD as well as to how the host’s immune system responds to the disease. This would mean studying and protecting the healthy gut microbiome would be useful as a preventative measure for susceptible patients, such as those with strong family histories and genetic disposition to the disease. It may also be utilised in patients currently having IBD. There may not be enough evidence to suggest it can cure those with IBD, but the evidence seems to show that reducing dysbiosis and maintaining a healthy balance of gut bacteria will help reduce exaggerated immune responses, thus reducing flares and maintaining remission.

Haneishi et al., 2023 [7] conducted a cohort study that showed patients and mice with IBD displayed different gut microbiota composition than non-diseased individuals. Jacobs et al., 2016 [8] found the imbalance of the gut microbiota to be associated with the cause of IBD. These could imply that interventions that target the microbiota, such as probiotics or faecal microbiota transplantation (FMT), could be helpful in treating IBD symptoms.

### Altered Tryptophan Metabolism in IBD as a Biomarker

Altered tryptophan (Trp) metabolism is increasingly recognised as a significant factor in the pathogenesis of IBD. Trp is an essential amino acid that has a role in gut microbiota homeostasis and is affected by many factors: probiotics, stress, ageing, and diseases including IBD [9].

Trp is a substrate for protein synthesis and is primarily metabolised via two main pathways: the kynurenine pathway (KP) and the kynurenic acid pathway (KA). Bacterial Trp metabolism differs slightly as bacteria can directly utilize Trp, but it is difficult in practice to identify which metabolites are produced, and further research would be needed to clarify this beyond the molecular level, thus making it impactful in clinical practice.

IBD is, however, associated with altered host and bacterial Trp metabolites. The plasma levels of kynurenine and KA are increased [10], whilst plasma Trp is decreased in patients with IBD [9]. It is thought the increased levels of circulating pro-inflammatory cytokines induce Trp catabolism. In animal studies, bacterial Trp metabolites are suggested to be significantly reduced in hosts with IBD, thus implying their beneficial role and potential anti-inflammatory properties. In humans with IBD, the low bacterial metabolite levels in faeces suggest that reduced Trp metabolism may be implicated in the aetiology of IBD [11]. Given the suggested relationship between Trp and its metabolites and the gut microbiome and host immune response, further research into potential therapies targeting this is warranted [9].

It could serve as a next-line treatment, broaden remission possibilities in IBD patients, and serve to potentially reduce the use of surgery to treat IBD patients.

## 3. Microbiota and the Role of Short-Chain Fatty Acids (SCFAs) in IBD

Certain species are responsible for the production of SCFAs. These SCFAs play an essential role with their anti-inflammatory properties and enhancement of the gut immune function. They also aid in the differentiation of regulatory T-cells into Th17 and Treg cells [12,13]. “Inflammatory reaction driven by Th cells protects the host from the detrimental pathogens”; however, “excessive activation of Th cells is associated with intestinal inflammation” [6]. The balance between pro-inflammatory Th17 and anti-inflammatory Treg cells is important in maintaining gut stability, which is directly affected by the content and diversity of the gut microbiota.

In the 1990s, a number of studies looked at a potential role for topical SCFA (butyric acid) treatment in distal ulcerative colitis through enemas or irrigation. Although trends towards beneficial effects were seen, there were no studies showing statistically significant results, and academic interest in the subject waned.

## 4. Prebiotics and Probiotics in IBD Treatment

Other dietary components that can affect the gut microbiome include prebiotics, postbiotics, and combinations of these known as synbiotics. Prebiotics are fermentable fibres found in foods, and probiotics are live, non-pathogenic microorganisms also implicated in the alteration of the gut microbiome for the benefit of the host. Synbiotics are a combination of the two and are thought to exert a synergistic effect [14]. 

Prebiotics and probiotics are thought to modulate the host immune response and improve intestinal barrier function and nutritional absorption, outcompeting pathogens for nutrition, adhesion to the gut lining, and production of antimicrobial substances [15].

Prebiotics are divided into groups based on the composition of sugars/carbohydrates. These naturally occurring compounds are found in many different foods but not in large concentrations [16]. They can be synthesised to make up for this, and, given their positive profile and therapeutic potential, it would be reasonable to assume they would be beneficial in IBD (Table 1).

Paudel et al. [17] commented that a diet deprived of fibre would enhance mucus-eroding microbiota and, thus, leave the host susceptible to opportunistic intestinal microbes. At the very least, a lack of dietary fibre is detrimental to the healthy gut microbiome. They also commented that other studies showed that certain fibres offered protection against colitis [16]. Rau et al. [14] noted that, in some studies, prebiotics, although safe, did have some side effects such as bloating and flatulence. They also noted that caution should be exercised when recommending increased-fibre diets to patients with stricturing Crohn’s disease; increased fibre is not necessarily a blanket treatment for all IBD. Whilst there is still scope for further research into prebiotic safety and efficacy, there exist some data to support the use of high-fibre diets, such as plant-based ones, to reduce symptoms in IBD [18,19,20].

Probiotics can also be found in fermented foods, and some have recently been marketed in the form of pills or capsules. They typically contain one or more microbial strains [12]. Rau et al. [14] noted varying outcomes of probiotics in IBD; they noted mixed reviews, with some studies showing ineffective induction of remission in UC. Some studies such as a systematic review and meta-analysis by Kaur et al. showed promising results [21], where probiotics were effective in “the induction of clinical remission in UC” and that “there were no overall differences in minor or serious adverse events when comparing probiotics with placebo”. These studies, however, seemed to mainly comment on the effects of probiotics in UC. Much like prebiotics, there is more evidence required regarding the safety and efficacy of probiotics, especially in CD, in which there has been limited research thus far, before they can be applied to regular clinical practice. A Cochrane review [22] on the treatment of pouchitis following ileal pouch–anal anastomosis suggested a beneficial trend with probiotics in acute pouchitis, but robust data are still lacking.

Synbiotics in IBD also have limited research, but their potential is not to be ignored. Given the potential benefit of both prebiotics and probiotics, a synergistic combination of the two could prove useful in overcoming the current limitations in research and prove to be a safe and efficacious future treatment for IBD.

## 5. Faecal Microbiota Transplantation

One novel therapy that is utilised in the modulation of gut microbiome is faecal microbiota transplantation (FMT). This intervention has the potential to restore the balance in the gut microbiome between potentially pathogenic bacteria and beneficial strains. There is also the possibility of the transplantation of damaged microbiota from a healthy donor to an affected recipient [23].

FMT’s exact mechanism of action is not fully understood. However, it is believed that, if successful, then the host’s response to the transplant will lead to the promotion of anti-inflammatory metabolites such as SCFAs, suppressing inflammatory metabolites and reducing overall inflammation and damage to the bowel [23]. If the correct beneficial species can be identified in the donor, then FMT has the potential to become a specific treatment for an individual recipient or become a part of their personalised treatment plan.

FMT has been very successful in treating certain gastrointestinal infections, such as recurrent Clostridium difficile infection, which can cause pseudomembranous colitis.

There are many factors that determine the success of FMT in IBD. There should be careful donor and recipient screening and stringent consideration of the storage and preparation of the FMT; treatment before FMT, e.g., drug therapy or bowel preparation; route of administration; dose; and number of sessions of FMT [23,24]. Some factors identified in this systematic review of 25 studies showed these recipient factors to be associated with predictors of success when carrying out FMT: young age, less severe disease, shorter duration of disease, microbiome containing higher faecal species richness, greater abundance of *Candida,* and similarity to donor profile.

Some studies show that FMT works less well in UC than in Crohn’s disease (CD). There are some reasons why it is more consistent in patients with UC than in those with CD; it seems to stem from the site of disease, as the microbial diversity of ileal and ilocolonic CD is significantly different from that of healthy controls, whilst colonic CD and UC were closer to each other [24,25].

Whilst FMT is generally safe, there have been some adverse effects reported in some trials in the younger and adolescent population with IBD. These effects include diarrhoea, abdominal cramps, bloating, and pain. These gastrointestinal symptoms are self-limiting and transient [26]. There have even been fatalities in up to 1.4% of FMT sessions [27,28].

Whilst these percentages are low, the safety of FMT is not guaranteed. These studies show that there is a need for the strict selection of patients as well as the regime they undergo as recommended by the European faecal transplant group [29].

Some upcoming advances are in the novel delivery of FMT such as colonic transendoscopic tube insertion for whole colonic treatment [30]. The tube insertions for this study were successful in all 54 patients; 98.1% of the patients were satisfied with this procedure. No discomfort or adverse events were seen. Patients were excluded if they had severe bowel lesions including stenosis, complex fistulas, or perianal lesions. This is an important consideration when selecting appropriate patients for therapy and when predicting how impactful specific therapies can be. These patients also had mesalazine delivered through their endoscopic tube, so the efficacy of FMT on its own for IBD patients is yet to be determined. Nevertheless, FMT holds promise for the future as a potential therapeutic target in IBD.

### 5.1. Microbiota Restoration Therapy (MRT)

FMT has, as demonstrated above, some potential side effects due to possibly unknown factors within the transplant. Another idea has focused on more defined/targeted microbial population delivery to the colon. This is known as microbiota restoration therapy (MRT). There has been promise with regards to Clostridium difficile treatment [31], although there is still more research to be carried out to solidify its therapeutic efficacy. The aim of MRT is to restore healthy gut microbiota to counter the dysbiosis in IBD. In the U.S., there has been Food and Drug administration (FDA) approval for the use of microbiome treatment containing live bacteria of qualified donor stool. This is a potentially huge step forward in the research on MRT. With this approval, its use can be studied with more reassurance and fewer administrative issues when setting up the research protocol.

### 5.2. Modification of Host–Microbe Interactions

The host–microbe chemical interactions in normal, healthy subjects contribute to normal gut health. They are responsible for the maintenance of gut barrier function, mucosal integrity, and clearance of pathogens. While FMT is one approach to restoring healthy gut microbiota interactions, another potential method is to modulate or modify host–microbe metabolic interactions.

Various biotechnological companies have examined these interactions and developed certain therapeutic targets and are studying their utility in IBD patients.

An experimental drug, EB8018, has been developed with the aim of blocking invasive *E. Coli* species’ adherence onto intestinal epithelial cells and is currently in phase 1b trials. Other compounds still in their preclinical phase of study include SG-2-0776, which is being researched in a preclinical model of IBD [17,32].

The compounds SYMB-104 and SYMB-202 show promise in increasing the levels of Treg cells and reducing overall levels of intestinal inflammation. This shows promise for a system-targeted approach in personalising treatment for IBD patients. If it can specifically target colonic inflammation whilst maintaining low immunogenicity, then we may see a more effective and lasting treatment on the horizon. And, for patients, this translates to effective medical treatment, greater variety of treatment, more room for escalation without surgery, and ultimately living with their disease symptom free in remission [32].

## 6. Nanoparticle Gene Delivery

One other novel advancement for the delivery of treatment in IBD is the development of oral nanomedicines, which have been shown to directly affect and restore damaged intestinal epithelium. Nanoparticles are defined as particles between 1 and 100 nanometres in size. There are 1,000,000 nanometres in 1 millimetre. Some of these studies have shown promising therapeutic benefit in mouse large and small bowel samples. These 3-D tissue culture models served as a safe predictive base for disease sequalae and response to novel therapies such as nanomedicines [33].

These 3-d models showed strong anti-inflammatory responses and the ability to target very specific parts of the inflammatory process, in recent years, even being able to modulate mRNA expression selectively [34].

Research shows problems with gene expression in many pathologies including inflammatory bowel disease. Veiga et al. [34] has shown ways to overcome some of the technological challenges when using modified messenger RNA (mmRNA), such as the ability to successfully transcribe large amounts of mRNA in vitro, instability of the mRNA in vivo, the amount of immune response the host has in response to the therapy, and the ability to target specific cells.

This shows exciting promise for the future of disease therapy and the application of mRNA-related therapies. There is potential to introduce IL-10, a predominantly anti-inflammatory cytokine associated with the regulation of IBD, mRNA as a novel therapy. The potential benefit of mRNA therapies also lies in their lower risk for genomic integration compared with their DNA-based treatment counterparts.

Veiga et al. [34] designed lipid nanoparticles (LNPs) to overcome potential shortcomings and to avoid using other vectors such as viruses, which may have hindered long-term therapy options. LNPs would prevent the degradation of RNA molecules and reduce the host immune response to them, which is ideal. It is also a feasible and replicable preparation method that allows a more standardised approach in its adoption as a novel treatment [35]. They used transmission electron microscopy to ascertain the efficiency of LNP encapsulation, which was ~100% [36,37].

They were also able to add targeting antibodies for the cell-specific delivery of mmRNA, and this method did not hamper mRNA encapsulation. Therefore, the therapy should be specific and only target the colon.

Safety and sustainability often go hand in hand when describing therapies. So, in order to check that the novel therapies are not just novel and are of potential use to humans, we must look at how specific such therapy is and whether there are any potential areas for development especially with systemic inflammatory reactions.

To assess the immunogenicity of their LNPs, splenic inflammatory cytokines were analysed via ELISA as a marker of systemic inflammation, and these were found to be in lower amounts than IL-10 levels in the colon and in control groups. This shows us that they targeted the specific expression of IL-10 in colonic mice models, with induced inflammatory bowel disease, without the systemic involvement of other organs. Pathological disease signs were also used as a marker including weight loss, erythema, and inflammatory infiltration of the colon. These were also found to be significantly reduced in the treated mice, and the control groups showed minimal reduction in these signs; thus, proving its therapeutic potential and safety to trial in human studies. The pro-inflammatory markers they looked at originated from the spleen, and the mice were sacrificed to look at their colonic histology after day 10. So, whilst there are very promising human implications for this treatment, there is scope for further analysis on the long-term safety and other systemic markers.

## 7. Conclusions

The research into the diagnostics and novel therapies available on the horizon for humans is very promising. Animal studies have shown potentially transferrable and safe therapies that can target specific sites of inflammation. Modulating the inflammatory response is a powerful therapy with what is shown to be a reasonably safe profile to build further research on. If properly and successfully developed, it would have life changing potential for refractory disease and provide patients with therapies associated with fewer side effects as compared to surgery. There is also potential for those with other inflammatory conditions to benefit from these innovative modes of treatment. The implications mentioned in this review for targeted gene therapy, effective treatment with low immunogenicity, and a safe side effect profile can be adapted to other conditions [38].

## Figures and Tables

**Figure 1 biomedicines-13-00716-f001:**
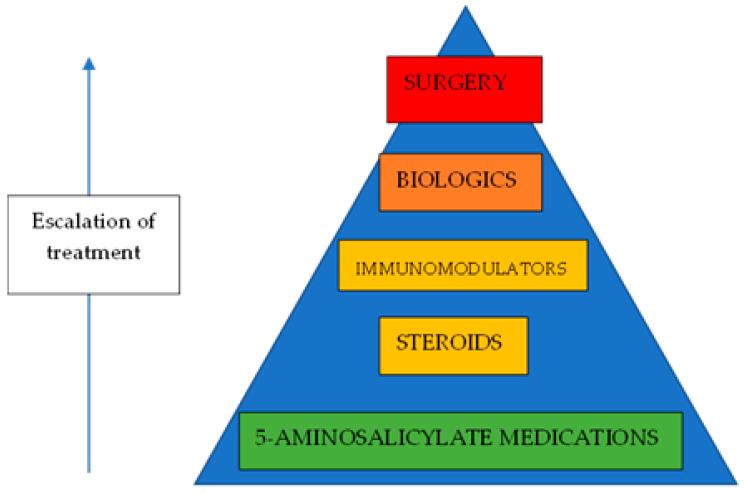
Treatment of IBD.

**Figure 2 biomedicines-13-00716-f002:**
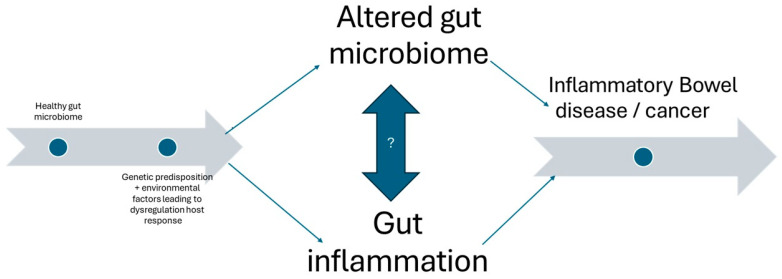
Overview of the relationship between altered gut microbiome and gut inflammation showing potential causative factors as well as serious sequalae.

**Table 1 biomedicines-13-00716-t001:** Advantages and limitations of probiotics and prebiotics.

Probiotics Pros	Prebiotics Pros	Probiotic Limitations	Prebiotic Limitations
Symptom relief	Anti-inflammatory	Strain and condition specific	Worsening of some symptoms
Enhancement of immunity	Reverses dysbiosis	Not universally effective	Limited evidence base
		Variable quality	
		Risk in immunocompromised	

## Data Availability

No new data were created.

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
