# Peer review of "Gut Microbial Targets in Inflammatory Bowel Disease: Current Position and Future Developments"

_biomedicines, 2025, doi:10.3390/biomedicines13030716_

Round 1

Reviewer 1 Report

Comments and Suggestions for Authors

Dear authors,

The manuscript describes the gut microbial targets in Inflammatory Bowel Disease (IBD), including their current position and future perspectives. It presents a solid point of view, here are some comments.

  1. Change the sentence in line 35-36 to "CD cannot be cured with the best hope of achieving remission."
  2. change the "food" in line 119 to "foods". 
  3. Intestinal microbes are quite different from gut microbiota, please check it out, particularly in line 131. 
  4. I recommend to conclude IBD treatment in a figure.
Comments on the Quality of English Language

english is fine 

Author Response

Comments 1: Change the sentence in line 35-36 to "CD cannot be cured with the best hope of achieving remission."

Response 1: Thank you for pointing this out. We agree with your comments and have changed this section of text

Comments 2: change the "food" in line 119 to "foods". 

Response 2:  Thank you, this has also been amended  as per your comments.

Comments 3: Intestinal microbes are quite different from gut microbiota, please check it out, particularly in line 131. 

Response 3: Thank you for pointing this out, we have changed the wording back to gut microbiota

Comments 4: I recommend to conclude IBD treatment in a figure.

Response 4:  Thank you, we have added a figure to summarise this.

Reviewer 2 Report

Comments and Suggestions for Authors

                       Report of the Manuscript ID biomedicines-3494980

The article submitted for our consideration is of considerable interest. However, we note that the authors should make certain corrections in order to improve the scientific quality of the manuscript review.

Introduction section

The introduction as a whole needs considerable improvement. The study problem, i.e. the issue, is well defined. However, we believe that a good introduction always ends with clear objectives for the study. In this case, it's not very clear. In the introduction, we note that the main research question does not appear, and also that the research hypotheses that stem from the main research question are absent. The authors should revise this in view of their importance. So the whole introduction needs to be improved.

Methodology section

In the methodology section, the causes of gut microbial targets in inflammatory bowel disease should appear. For example, the authors should tell us about stomach acidity and antibiotic misuse, which could influence intestinal inflammation. In short, the authors should discuss the multiple factors involved in the pathogenesis of IBD, such as genetic, environmental, infectious and immunological factors, among which intestinal inflammation and immune dysfunction play an important role. Also, the authors will have to revise and enrich the methodology section speaking of species bacteria species and functional pathways involved in the protection and pathogenicity of inflammatory bowel disease.

Author Response

Comments 1: 

Introduction section

The introduction as a whole needs considerable improvement. The study problem, i.e. the issue, is well defined. However, we believe that a good introduction always ends with clear objectives for the study. In this case, it's not very clear. In the introduction, we note that the main research question does not appear, and also that the research hypotheses that stem from the main research question are absent. The authors should revise this in view of their importance. So the whole introduction needs to be improved.

Response 1: Thank you very much for taking the time to review our paper and giving such in depth feedback. We agree with your comments and have revised the introduction below:

An evolving area of scientific interest involves the gut microbiome – which is the community of micro-organisms living in the human digestive tract. Specifically, there exists a complex interplay between gut microbiome species and cells of the gastrointestinal tract – and this is implicated in both maintenance of physiological health but also the development of disease. An imbalance in the gut microbiome – where certain deleterious species of microorganisms are increased relative to the more “favourable” species – is known as gut dysbiosis. In fact, it has been shown that IBD bacterial families that exist in abundance do not co-exist together unlike in healthy individuals (ref) . There is also emerging data to suggest that gut dysbiosis is associated with not just IBD but its complications such as colorectal cancer [1,2]

Modulation of the gut microbiome could reduce inflammation and thus improve symptoms, maintain remission and reduce relapses in IBD without the dependence of medications with significant side effects. To draw a parallel, cancer treatment is evolving rapidly with tailoring of therapies towards individual genetic mutations and “individualisation” of treatment. It is possible that the key to similar advances in IBD treatment lies within gut microbial therapeutic targets.

In this article we aim to review current and emerging gut microbial targets in IBD, and specifically to elucidate the future developments most likely to impact practice.

Comments 2: 

Methodology section

In the methodology section, the causes of gut microbial targets in inflammatory bowel disease should appear. For example, the authors should tell us about stomach acidity and antibiotic misuse, which could influence intestinal inflammation. In short, the authors should discuss the multiple factors involved in the pathogenesis of IBD, such as genetic, environmental, infectious and immunological factors, among which intestinal inflammation and immune dysfunction play an important role. Also, the authors will have to revise and enrich the methodology section speaking of species bacteria species and functional pathways involved in the protection and pathogenicity of inflammatory bowel disease.

Response 2: 

Thank you again for these comments and in depth feedback. 

We accept the reviewers point but feel that straying into gastric acidity and antibiotic misuse is beyond the scope of this article. Similarly the pathogenesis as well as immune dysfunction in IBD is out of scope and worthy of review in itself. We have referenced a recent paper in this area.

PMID: 32076145 DOI: 10.1038/s41575-019-0258-z

Reviewer 3 Report

Comments and Suggestions for Authors

The review “Gut Microbial Targets in Inflammatory Bowel Disease: Current Position and Future Developments” by Sivakumar and colleagues comprehensively discusses the heterogenous nature of the inflammatory bowel disease and emphasizes on the need to explore gut microbiome for designing improved targeting and therapeutics. The review emphasizes on the need to investigate the disease more deeply as it presents with a lot of challenges and inferior quality of life.

The review is well written however, it can be improved in quality by a minor revision. Some of the comments are as follows:

1. The review lacks flow at several places and appears like a simple compilation of study findings, e.g page 3, line 81-102. It can be improved by a rewriting.

2. A table comparing advantages and limitations in recommending Prebiotics and probiotics.

3. The understanding and simplicity of the review will be highly benefitted by the presence of stand-alone figures. Currently, the review lacks figures. Fig.1 doesn’t add much value.

4. It would be highly advantageous to discuss and include findings from recent clinical trials going in this area.

Author Response

Comments 1: The review lacks flow at several places and appears like a simple compilation of study findings, e.g page 3, line 81-102. It can be improved by a rewriting.

Response 1 : Thank you for pointing this out. We agree with this comment and have reflected on the fluency of our writing especially this section mentioned. We have re-written this section. It is now lines 117-138.

Comments 2: A table comparing advantages and limitations in recommending Prebiotics and probiotics.

Response 2: Thank you very much, we agree and have added it as table 1 at the end of section 4.

Comments 3: The understanding and simplicity of the review will be highly benefitted by the presence of stand-alone figures. Currently, the review lacks figures. Fig.1 doesn’t add much value.

Response 3: Thank you very much for this comment. We agree with this comment and have tried to space out our added figures and diagrams to improve the overall flow and accessibility of the article

Comments 4: It would be highly advantageous to discuss and include findings from recent clinical trials going in this area.

Response 4: Thank you for this comment. We agree with it and have discussed some recent trials in heading 5.2.

Round 2

Reviewer 2 Report

Comments and Suggestions for Authors

We think that the authors have made some fairly significant improvements. Therefore, the manuscript article can be accepted in its present form.